

# Fault distance-based approach in thermal anomaly detection before strong Earthquakes

Arash Karimi Zarchi[1], Mohammad Reza Saradjian Maralan[1]

[1]Remote Sensing Department, School of Surveying and Geospatial Engineering, Collage of Engineering, University of Tehran, Tehran, Iran

*Correspondence to*: Arash Karimi Zarchi (Arash.karimi93@yahoo.com)

**ABSTRACT**

The recent scientific studies in the context of earthquake precursors reveal some processes connected to seismic
activity including thermal anomaly before earthquakes which is a great help for making a better decision regarding this disastrous phenomenon and reducing its casualty to a minimum. This paper represents a method for grouping the proper input data for different thermal anomaly detection methods using the land surface temperature (LST) mean in multiple distances from the corresponding fault during the 40 days (i.e. 30 days before and 10 days after impending earthquake) of investigation. Six strong earthquakes with Ms > 6 that have
occurred in Iran have been investigated in this study. We used two different approaches for detecting thermal anomalies. They are mean-standard deviation method also known as standard method and interquartile method which is similar to the first method but uses different parameters as input. Most of the studies have considered thermal anomalies around the known epicentre locations where the investigation can only be performed after the earthquake. This study is using fault distance-based approach in predicting the earthquake regarding the location
of the faults as the potential area. This could be considered as an important step towards actual prediction of earthquake's time and intensity. Results show that the proposed input data produces less false alarms in each of the thermal anomaly detection methods compared to the ordinary input data making this method much more accurate and stable considering the easy accessibility of thermal data and their less complicated algorithms for processing. In the final step, the detected anomalies are used for estimating earthquake intensity using Artificial
Neural Network (ANN). The results show that estimated intensities of most earthquakes are very close to the actual intensities. Since the location of the active faults are known a priori, using fault distance-based approach may be regarded as a superior method in predicting the impending earthquakes for vulnerable faults. In spite of the previous investigations that the studies were only possible aftermath, the fault distance-based approach can be used as a tool for future unknown earthquakes prediction. However, it is recommended to use thermal
anomaly detection as an initial process to be jointly used with other precursors to reduce the number of investigations that require more complicated algorithms and data processing.

**KEYWORDS:** Earthquake Precursor, Thermal Anomaly, LST, Fault, ANN



## 1. INTRODUCTION

Earthquake is one of the most difficult phenomena to predict and hence one of the most destructive natural calamities, capable of causing lots of instant loss of lives and property (Console et al., 2002). Earthquake is the result of surging tectonic stress and its release caused by tectonic movement in fault zones (Geller et al., 1997). There have been a number of studies regarding the existing precursors for this natural disaster. These studies indicate some of the precursors such as water temperature, water level change, flow rate, magnetic field, electric

field, soil/air temperature, relative humidity, ionospheres fluctuations, surface deformations, and land surface temperature (LST) anomalies (Cicerone et al., 2009; Hayakawa et al., 2000; Wyss, 1997; Yao, 2007).

Although each of these precursors has an important and individual role in estimating the earthquake parameters, the thermal anomaly precursor is one of the most useful ones due to its physical relevance to the nature of the earthquake. There are mainly five hypotheses describing the physical basis behind thermal anomaly related to

45 seismic activities: 1) Earth degassing mechanism, 2) Groundwater anomaly, 3) Frictional heating, 4) Seismo-ionosphere coupling, and 5) p-hole activation mechanism (Cicerone et al., 2009; Freund, 2011; Qiang, 1999; Roeloffs, 1988). Each of these concepts will either directly increase the land surface temperature or they will increase the near-surface air temperature that would eventually cause the change in surface temperature depending on their effect (Saraf et al., 2009; Surkov et al., 2006; Tramutoli et al., 2001; Verma and Bansal,

2012).

The idea that thermal anomalies may be related to seismic activity was first put into application in the early eighties. In 1990, researchers detected thermal anomalies prior to an earthquake (Tronin, 1996). These researchers were among the first people who suggested using the thermal infrared anomalies for a better understanding of the earthquake phenomena. Tramutoli (1998) used Robust AVHRR Approach (RAT) method

for detecting thermal anomalies before the earthquakes using AVHRR data (Tramutoli, 1998). This method allowed Tramutoli to detect the thermal anomalies with better accuracy by separating the natural anomalies like the seasonal changes from the anomalies that were related to seismic activities (Tramutoli et al., 2001). Later this method was adjusted for the rest of the remote sensing data and it is now known as Robust Satellite Approach or RST (Tramutoli et al., 2005).

Originally, in order to describe the relationship between land surface temperature and seismic activities AVHRR (Advanced Very High-Resolution Radiometer located onboard NOAA satellite) data were used (Qiang et al., 1997). Many studies like Saraf (2009) and Choudhury (2006) were done investigating several strong earthquakes using AVHRR data (Choudhury et al., 2006; Saraf et al., 2009). In a few cases, they used the Defense Meteorological Satellite Program (DMSP) in certain situations like areas with snow or cloud cover in

order to produce higher accuracy (Saraf and Choudhury, 2005). Although they did not explore the statistical parameters of the anomaly and relied only on visual interpretation, their study's widened spatial coverage required others to take an important step for future studies in this case. Rawat in 2011, investigated Ms 5.9 Vrancea (Romania) earthquake on October 27th, 2004 and showed LST rise of 5–10 °C within a week of the earthquake (Rawat et al., 2011).

In recent years, MODIS LST (Moderate Resolution Imaging Spectroradiometer) product has proven to be very useful as a direct input for several study cases and various anomaly detection methods. In 2004 Ouzounov and Freund were among the first researchers who used MODIS products as their input data for their studies (Ouzounov and Freund, 2004). They also investigated TIR anomalies and mid-IR emissions prior to an





earthquake. Another study conducted one of the most destructive earthquakes that happened in Gujarat (India)

on January 26th, 2001. The results show the appearance of an anomaly of 3–4 °C about 5 days before the earthquake (Ouzounov and Freund, 2004).

As mentioned, earthquake occurs as a result of releasing the built-up stress along the fault. Therefore, active faults are one of the most important factors contributing in the earthquake process. Although many studies have focused on the subject of the earthquake, only a few of them have investigated the changes that happen along

the fault zones regarding their temperature. In 2010, Wang and Manga investigated the groundwater change in a fault zone that would later cause a change in land surface temperature in multiple earthquakes (Wang and Manga, 2010). Later in 2019, Li and Shi investigated anomalies in Earth degassing mechanism and groundwater in 2008 Wenchuan Ms 8.0, 2013 Lushan Ms 7.0 and 2014 Kangding Ms 6.3 earthquakes near Xianshuihe fault zone (Li et al., 2019). Although these studies were not directly about the thermal anomalies, the results showed

releasing $CO_2$ and changes in the composition of groundwater that would later cause a change in land surface temperature.

Recently, many studies use the help of machine learning algorithms specially those involving Artificial Neural Network (ANN). ANN is a mathematic model and was adapted from human reactions and simulation of thinking processes to complicated problems. Using ANN, one can solve these problems without entering into

90 complex theories and topics (Adeli, 1999). In 2014, Akhoondzadeh investigated Saravan earthquake in Iran that occurred on April 16th, 2013 using ANN with the help of Particle Swarm Optimization (PSO) to increase the efficiency of thermal anomaly detection (Akhoondzadeh, 2014).

This paper presents a method of grouping input data for using two deferent anomaly detection methods. Most of the studies on thermal anomalies are only possible after the earthquake happens since they require the location

of the epicentre. The presented Fault distance-based approach can be a better method in predicting the earthquake, as it uses the known location of the fault, which is related to the impending earthquake. The conventional data selection uses only LST and time, while distance-based grouping of data proposed in this study would also take into consideration one of the very influential parts of the earthquake effect which is the relevant fault and the regions around it. This study also intends to show the accuracy of using the combination

of this assembled data and thermal anomaly detection results for estimating each earthquake's intensity using ANN.

## 2. STUDY AREAS AND DATASETS

### 2.1 Study sites

Table 1. The earthquakes' locations, Intensities and dates of six strong earthquakes (Ms>6) of Azgalah, Goharan,

Saravan, Shonbeh, Brujerd and Sari studied in this paper.

Table 1. The earthquakes' locations, Intensities and datesin this study

| Earthquake | Epicenter | Intensity | Date |
|---|---|---|---|
| Azgalah | 34.81 N  45.83 E | 7.3 | November 12, 2017 |
| Goharan | 26.52 N  57.76 E | 6.2 | May 11, 2013 |
| Saravan | 27.11 N  62.05 E | 7.8 | April 16, 2013 |
| Shonbeh | 28.48 N  51.58 E | 6.3 | April 9, 2013 |
| Brujerd | 34.57 N  48.79 E | 6.1 | March 31, 2006 |
| Sari | 36.27 N  51.57 E | 6.3 | May 28, 2004 |





### 2.2 Datasets

In this study, MODIS sensor daily land surface temperature product (MOD11A1) during forty days (thirty days before and ten days after the earthquakes) for each earthquake has been used. The MODIS daily LST and emissivity data are retrieved at 1km pixel size by the generalized split-window algorithm, which uses bands 31

and 32.

In addition, the relevant active fault was identified and its shape file was extracted depending how close it was to the location of each earthquake's epicentre.

### 3. METHODOLOGY

This paper presents a method of grouping input data for different thermal anomaly detection methods. This uses

the land surface temperature mean in multiple distances of 1 to 20 km from the corresponding fault during the forty days starting from 30 days before and 10 days after a given earthquake event. In order to generate the input data and use it in the anomaly detection algorithm, the following steps have been performed: pre-processing, fault distant map, and land surface temperature diagram. The data then are used in anomaly detection methods and Artificial Neural Network (ANN).

### 3.1 Pre-processing

The first step is to remove the natural and observational noise signals, which are due to changes in seasons, view angles and air density from the TIR data. By doing so, the remaining data would be mainly unmixed TIR anomaly data associated with increased seismic activity. In order to achieve this, a linear function was fitted to the LST of the previous year with no strong seismic activities and then was subtracted from the present year of

125 LST in which the earthquake had occurred.

### 3.2 Fault distant map

In order to use the fault in our process it is necessary to have an understanding of the corresponding fault and its surrounding areas of different distancing. Fault distant map is a map that its pixels represent values depending on how far they are from the fault. The closer the pixel to the fault, the lower its value and it will increase as we

get further from it. Figure 1 shows the example of Azgalah fault distant map.

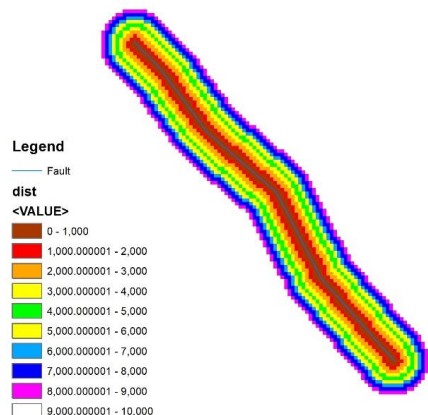

Figure 1 Fault distant map for Azgalah study case


### 3.3 Land surface temperature diagram

This paper presents a method that uses the temperature mean in different buffers with various radiuses (i.e. 1-20 km) around the related active fault during the period of investigation. As can be seen in Figure 2, the data is

135 shown by a 3D diagram, made by the LSTs mean in different radiuses around the related active fault in each day. This means that each pixel in this data which can be represented as a picture or 3D diagram show the LST mean in a certain radius buffer zone for a specific day. It should be noted that width of each buffer is only 1 km and R is the buffer radius (distance) from the related fault.

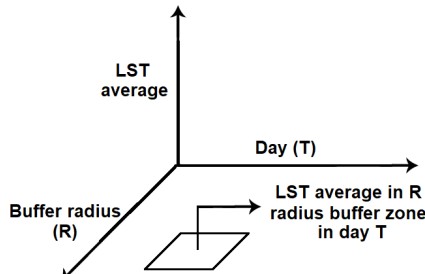

Figure 2. Schematic diagram of the 3D input data for different
anomaly detection methods

Later, these temperatures mean of each buffer is used as an input data to test various anomaly detection methods such as interquartile method and standard deviation method.

Instead of using conventional 2D-data, by using this 3D-data which include LST mean values in different buffers, time laps (days) to the earthquake events, and the distances from the fault, the anomaly detection methods will act more appropriately.

### 3.4 Anomaly detection methods

Two anomaly detection methods have been used in this study. The first one is simply the use of mean and standard deviation (Equation 1) of LST values (Akhoondzadeh, 2011) in each buffer zone.

$$x > \mu + k \times \sigma \qquad \text{Equation 1}$$

where $\mu$ is the mean value and $\sigma$ is the standard deviation value of the LSTs and $k$ is a coefficient around 1.6 but may slightly change for each study site. For each $x$ (*i.e.* LST), if the result of Equation 1 is true, it will be

regarded as an anomaly.

The second anomaly detection method uses a similar approach but instead of using mean and standard deviation, it uses median and interquartile range (Equation 2) (Saradjian and Akhoondzadeh, 2011) which is known as the Interquartile method.

$$x > M + k \times IQR \qquad \text{Equation 2}$$

where $M$ is median value, IQR is the interquartile range and $k$ is a coefficient around 1.3 but may slightly

change for each study site. Like the first method, if each $x$ (*i.e.* LST) is greater than $M + k \times IQR$, then the behaviour of the LST will be regarded as anomalous.





### 3.5 Artificial Neural Network (ANN)

Since ANN has shown to be one of the most reliable methods for solving many complicated problems, there are thousands of studies explaining its theoretical background (Cheng and Li, 2008; Chung et al., 2005; Ergu et al., 2014; Sahoo et al., 2016). Therefore we will only discuss a little about its basics and will focus on its details and results. ANN is a mathematical network model trained by using a specific set of data. This trained network can later be used to transform other sets of data to output (Nedic et al., 2014). Most ANNs consist of three different layers: input layer, which is a layer for initiating data, hidden layer, which could have multiple layers in it depending on the nature and complexity of the problem, and at last, output layer (Pradhan and Lee, 2009). Inside of each layer lays a number of neurons and nodes. Depending on how the network is trained, ANNs are divided into two categories: feedforward-propagation and back-propagation. Back-propagation ANNs are usually used in studies due to their better performance in various fields. One of the most common back-propagation ANNs is the multi-layer perceptron (MLP) network (Pradhan and Lee, 2010). Like many networks, MLP network consists of the above mentioned layers. Input data will be connected to the hidden layer using a number of weights and bias in each neuron thorough an activation function. Activation (transformation) function is chosen depending on the trial. By using a set of training data these weights and bias will be determined and later be optimized thorough a number of iterations (Nedic et al., 2014).

Since each earthquake under investigation happened in different region and time of the year, base temperature is different in each case. The different temperature between the detected anomaly pixel and its surrounding pixels in various buffers and days are the input data for training this network so that the different temperatures in each region do not cause a problem. Also all the temperature data used for training belongs to 30 days before the earthquake. In other words, the temperatures related to the 10 days after the earthquake in each case study are not used for training ANN.

### 4. RESULTS AND DISCUSSION

#### 4.1 Land surface temperature diagram

As mentioned, for each earthquake, the LSTs for each buffer zone was categorized and by using the LST mean of each buffer, its 3D-diagram was created (Figure **3**) and as input data used in thermal anomaly detection methods. Each pixel in these diagrams shows the LST mean (in Kelvin) in a certain radius buffer zone for a specific day. The red lines in Figure **3** show the day of the earthquake for each case. Due to using the proposed method for grouping the LST data, some of the anomalies can be seen even visually around the time of the earthquake. Although relying only on visual aspects isn't accurate enough, it can be used for better presenting and understanding the situation.




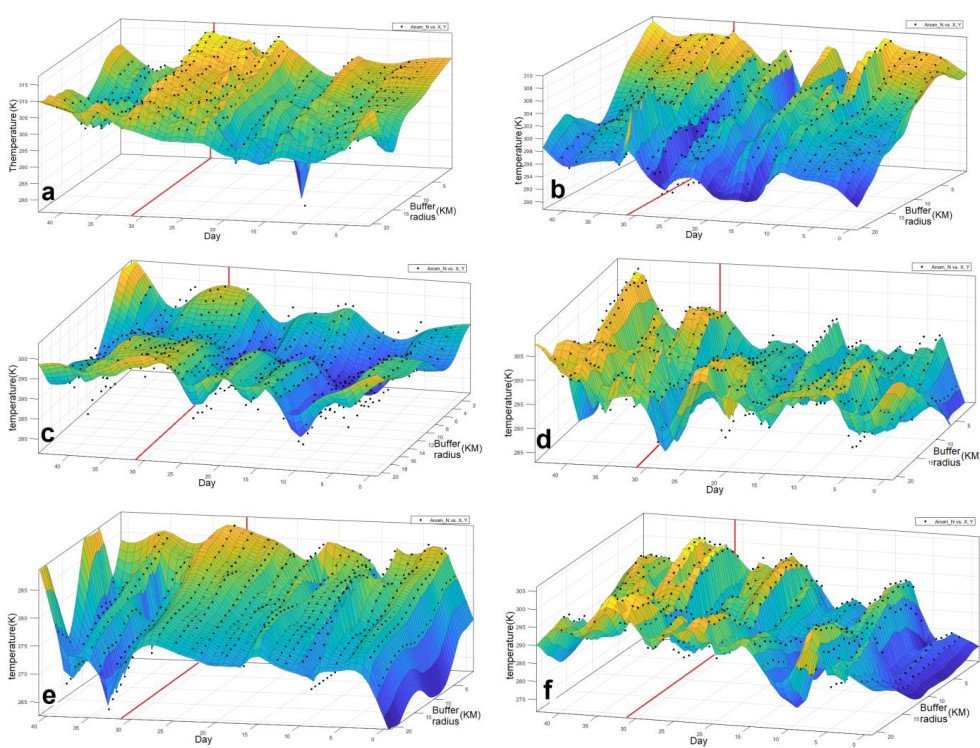

Figure 3: Land surface temperature diagram for a) Azgalah, b) Goharan, c) Saravan, d) Shonbeh, e) Brujerd, f) Sari case
studies where in each case, the earthquake day is 31th day

### 4.2 Detected thermal anomalies

In Figure 4 and Figure 5, the output of each anomaly detection method for each earthquake is shown. Results of each earthquake investigation show that the thermal anomaly is detectable in both of the anomaly detection methods mostly on the closest day to the earthquake regarding the closest buffer zone to the fault. These anomaly detection methods were used in other studies by using conventional data as input. Although they have detected some anomalies, their accuracy was always in question due to many false alarm anomalies detected along with the actual anomaly.

Results show that in Azgalah, Goharan, Saravan, Brujerd and Sari case studies, the anomalies detected by both methods are either on the day of the earthquake, the day before, the day after, or all of days mentioned. This difference is due to temporal proximity between the time of the imaging and the earthquake and earthquake's intensity. In Shonbeh case study, although a thermal anomaly was detected on the day of the earthquake, another slightly stronger anomaly was detected 8 days after that.

It should be noted that the anomalies detected in far distance buffers from the related fault are different for each earthquake (mostly in Saravan and Sari case studies) and do not have similar pattern. Moreover, since these pixels are far away from the related fault and epicentre, it cannot be said for certain that they are related to the earthquake. Therefore, these pixels were not considered as earthquake related anomaly and only anomalies in close distance buffers were used as earthquake related anomalies in ANN algorithms.

The difficulty of this method is in far distances, for example in buffers as far as 20 km radius from the fault, two pixels inside the buffer can be up to 80 km apart from each other, depending on the length of the fault itself. As a result, buffers with large radiuses could have pixels with various land covers and different temperatures. While limiting the buffer radius could shorten radiuses from the fault, it would make the area and diagram under investigation to become too small, causing the method to be less effective.

Changing the coefficient value ($k$) for each anomaly detection process affects the result. The higher the value of coefficient $k$ is, the higher the threshold for anomaly detection is set. This reduces the anomalies that can be detected while lowering the number of false alarm anomalies. On the other hand, increasing the coefficient value could result in omitting even the main anomaly that is related to the earthquake. Therefore, it is necessary to find an optimal value to increase the efficiency of each method. In this study, the coefficient value for standard method is around 1.6 and for interquartile method is around 1.3.

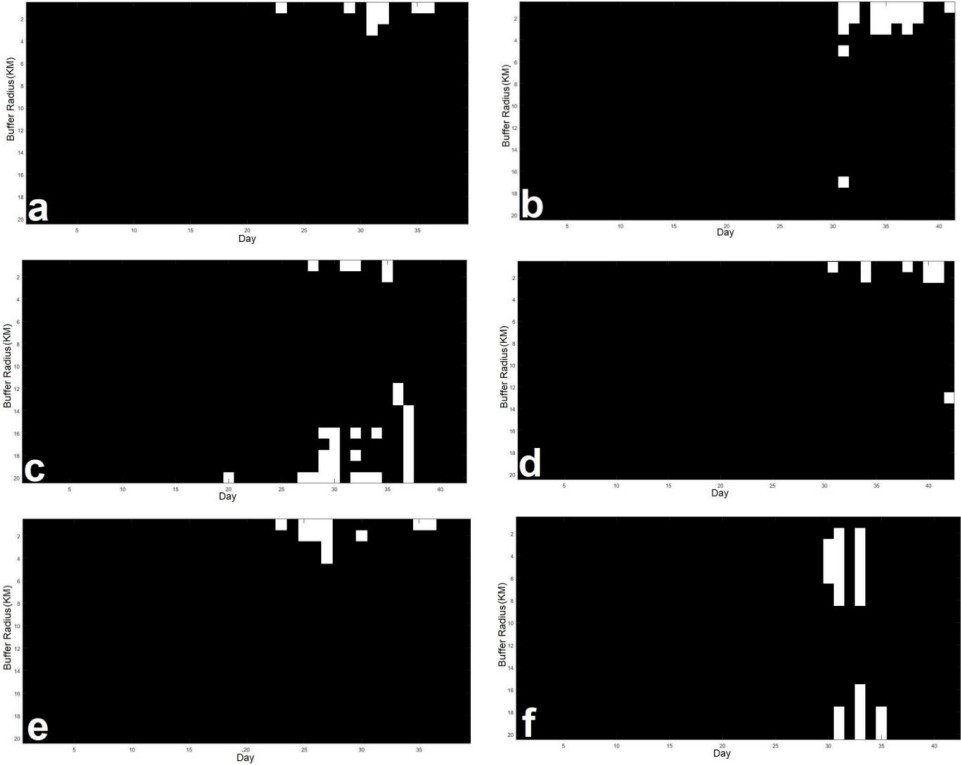

Figure 4 Simple anomaly detection method results for a) Azgalah, b) Goharan, c) Saravan, d) Shonbeh, e) Brujerd, f) Sari case studies where in each case, the earthquake day is 31th day

## 4.3 The impact of difference in Anomaly detection methods

The results show that both anomaly detection methods do find the thermal anomaly caused by seismic activities in each investigated earthquake. However, interquartile anomaly detection method has a slightly more specified outcome and less false alarm anomalies.

Figure 4 shows the results for a standard anomaly detection method. It indicates the anomalies detected around the time of the earthquake in the nearest buffer from the related fault. In Azgalah, Saravan and Brujerd cases, few anomalies are detected before the earthquake while in Goharan and Shonbeh cases few anomalies are detected after the earthquake. In Sari case, the earthquake related anomalies are detected on the day of the earthquake. However, the anomaly is not detected in the nearest buffer but in the 2-8 km buffer zones. In Azgalah, Saravan, Brujerd cases, some of the anomalies were detected around 6 days before the earthquake.

Although these anomalies are not as strong as the anomalies detected near the time of the earthquake, they seem to be related to some seismic activities rather than being a false alarm.

Results for interquartile anomaly detection method can be seen in Figure 5. Many anomalies detected by this method are related to the earthquake and found near the time of the earthquake in the closest buffer to the related fault with exception of Shonbah earthquake. As mentioned before, in Shonbah case study, another

thermal anomaly was detected beside the main anomaly, almost 8 days after the earthquake, which was even stronger than the anomaly related to the earthquake. Nevertheless, these results show that Interquartile anomaly detection method has more specified results and a better outcome for training ANN, compared to standard anomaly detection method.

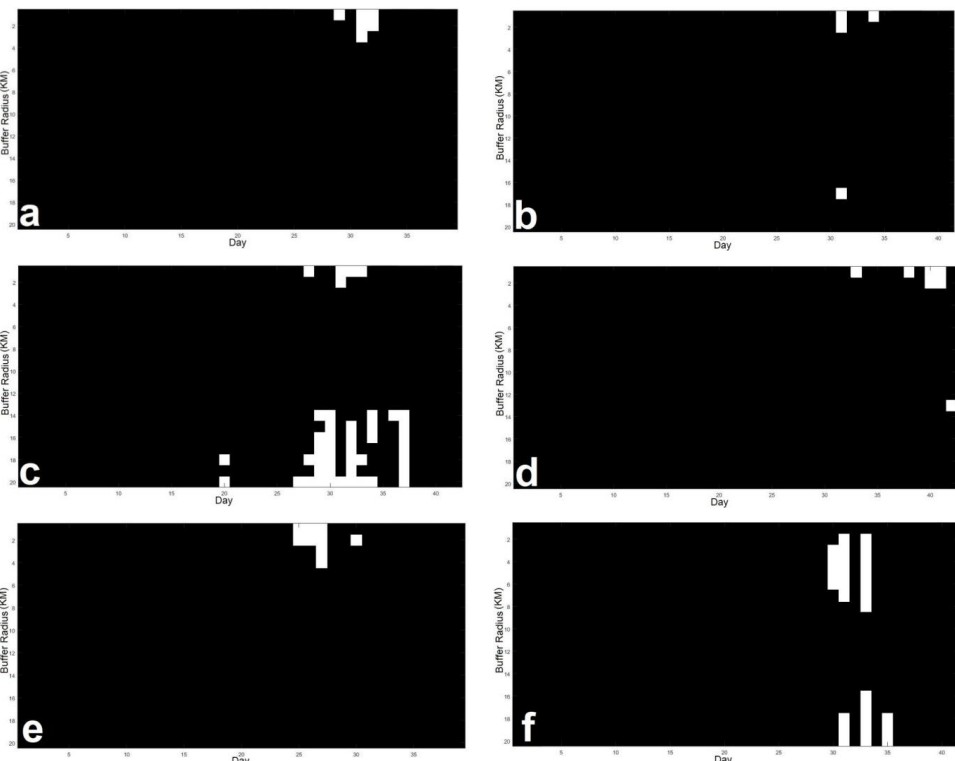

Figure 5 Interquartile anomaly detection method results for a) Azgalah, b) Goharan, c) Saravan, d) Shonbeh, e) Brujerd, f) Sari case studies where in each case, the earthquake day is 31th day





### 4.4 Artificial Neural Network (ANN) results

Since interquartile method created more precise inputs for training the ANN, the anomalies detected by this method were used. Table 2 shows the results of each earthquake's estimated intensity and its accuracy compared to its actual intensity. The results indicate that the best accuracy belongs to Azgalah and the one with least accuracy belongs to Sari case study. ANN results also show high correlations between thermal anomaly data and the earthquakes intensity.

Table 2 Estimated intensity for each earthquake using ANN

| Earthquake | Estimated Intensity | Error |
|---|---|---|
| Azgalah | 7.301 | 0.001 |
| Goharan | 6.302 | 0.102 |
| Saravan | 7.719 | 0.081 |
| Shonbeh | 6.29 | 0.01 |
| Brujerd | 6.28 | 0.02 |
| Sari | 6.245 | 0.145 |

Considering the limited number of investigated earthquakes, ANN did a great work by managing to sustain a good accuracy using various thermal data for each earthquake.

### 5. CONCLUSION

Thermal anomaly is indeed a significant precursor for strong earthquakes. The proposed method which includes analysis the anomalies with respect to the buffer zones in different distances relevant to faults, increases the accuracy dramatically. Two thermal anomaly detection methods were used for investigating each earthquake in this study. Although the outcome of each method is slightly different from another for each earthquake, interquartile method has better results compared to standard method. Nevertheless, they are both more accurate when anomaly detection algorithms use the proposed grouped inputs data instead of the ordinary data.

ANN results show that thermal anomaly data highly corresponds with earthquake intensity. Thus, the network was constructed properly, making the estimated results close to actual intensities. It is recommended to use more data related to more earthquakes and different locations for training ANN to improve the network accuracy.

However, it should be pointed out that thermal anomaly on its own is not quite sufficient for estimating the earthquake parameters and activities. It is highly recommended to use it as an initial and primary precursor for limiting the search area and then use other precursors, which require more complicated data and methods. Thermal anomaly precursors can also be used in combination with other simple precursors to get efficient and comprehensive results.

Many previous studies that investigated thermal anomalies, explored areas only around the epicentre. Methods used in such studies required the exact location of epicentre therefore they are only possible after happening of the earthquake. Since the location of the active faults are known a priori or can be identified by further investigations, using fault distance-based approach can be a superior method in predicting the impending earthquakes for vulnerable faults. In spite of the previous investigations that the studies were only possible aftermath, the fault distance-based approach can be used as a tool for future unknown earthquakes prediction.



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
