# Peer review of "Fault distance-based approach in thermal anomaly detection before strong Earthquakes"

_Natural Hazards and Earth System Sciences, 2020_

## Referee Comment (RC1) · Anonymous Referee #1 · 7 Jan 2021

***Review to: nhess-2020-391 paper, "Fault distance-based approach in thermal anomaly detection before strong Earthquakes", by Arash Karimi Zarchi and Mohammad Reza Saradjian Maralan***

The paper deals with a novel method of using thermal anomalies in LST records to investigate their correlation with strong Earthquakes and to assess their possible use for EQ forecasting. The paper is well written and organized but I cannot recommend publication in its present form, according to my comments here in the following.

**General comments**

Many papers claiming any relationships between physical/environmental parameters (as possible precursors) and earthquakes typically deals with the occurrence of anomalies before (or after) the quake. Only a few also look at the possible existence of such anomalous patters even in unperturbed periods (i.e. when no EQ occurred). The present paper suffers from the same strong limitation: it investigates 6 intense EQs occurred in the study area by means of land surface temperature (LST) anomaly analysis, demonstrating the occurrence of LST anomalies before or after the EQ, but it doesn't verify if similar thermal patterns occurred even when no EQ occurred in the area. This is, in my opinion, the main weakness of the paper that needs to be improved, including, at least, a full "falsification/confutation" analysis to be considered for publication.

Other general comments refer to some hypotheses, at the basis of the rationale of the study, that are poorly scientifically based and should be carefully handled.

1) Fault identification: often, a "fault systems" rather than a single fault, is activated during strong EQs. How the method can handle this? Not clear. In addition, the identification and selection of the fault is, even in this work, carried out aftermath, when the epicentre position is well known: how can manage this in terms of EQ forecast?

2) LST anomaly identification: to remove natural/observational noise, authors consider "…a previous year with no strong seismic activities…" and subtract this "linear function" from the data. In my opinion, 1 year is not enough to be considered as representative of the actual "normal" conditions of the area. A single year, in fact, can be affected by other (e.g. meteorological) forcing factors, limiting its representativeness for a "normal" year and significantly impacting in thermal anomaly identification.

3) coefficients k: to detect a thermal anomaly, authors apply two different methods, both requiring a threshold test based on a "k" coefficient that seems determined in a totally arbitrary way. K values determination, as well as their possible variability and dependence on different environmental/observational/geographical conditions should be better justified and assessed. Additionally, it is not completely clear how the average and sigma are computed.

4) Buffer radius: authors consider LST mean computed in buffer radius from 1 to 20 km. Why limiting R < 20km? It is well known that the "preparation zone" of a strong EQ could be as large as the Dobrovolsky theory (Dobrovolsky et al. 1979), i.e $R = 10^{0,43M}$ (with M= magnitude). In particular, for M=6 a radius of about 400 km can be expected for the EQ preparation zone. Moreover, if thermal anomalies are related to fault degassing, gases (e.g. $CO_2$) might be spread in large areas, also depending on meteorological conditions (e.g. wind intensity and direction) and/or local topography. Therefore, the limitation at 20 km should be better explained and scientifically justified.

5) How ANN works to estimate EQ intensity? Have the authors trained the network using the same dataset they used for results or an independent one?

**Specific comments**

- Figure 3: what do the black dots mean?
- It is not clear how the fault distance map is used in the process. Please clarify.
- Not clear how the LST mean is computed in buffer radius. In particular, please better explain the following sentence: "*It should be noted that width of each buffer is only 1 km and R is the buffer radius (distance) from the related fault.*"
- Lines 204-207: authors assert that anomaly far from the epicentre are not used for ANN. Thus, authors are using epicentre information for filtering data, but epicentre position is only available after EQ occurred, so this study appears as a retrospective analysis as well, and cannot be used for forecasting.
- Lines 230-231: "*Although these anomalies are not as strong as the anomalies detected near the time of the earthquake, they seem to be related to some seismic activities rather than being a false alarm*": how authors can assert this? What is the scientific basis for asserting this?
- Lines 233-234: …" *Many anomalies detected by this method are related to the earthquake…*". Again, how authors can affirm this?
- Lines 244-245: "*ANN results also show high correlations between thermal anomaly data and 245 the earthquakes intensity.*" What exactly does this sentence mean?

**Cited reference**

Dobrovolsky I P, Zubkov S I and Myachkin V I, 1979 Estimation of the size of the earthquake preparation zones Pure Appl. Geophys. 117 1025–44.

---

## Author Comment (AC1) · 27 Jan 2021

We appreciate the time and effort that you have dedicated to providing your valuable feedback and your insightful comments on our manuscript. We have been able to incorporate changes to reflect most of the suggestions you have provided and will be accessible in our next uploading manuscript.

Before we get to the comments, we believe it is better to explain how this method can be used in predicting the earthquake:

Consider 5 of the earthquakes have already occurred and the 6th earthquake has not

happened yet. Using the information regarding these 5 earthquakes we could find the corresponding fault since the locations of the epicentres are known for these 5 earthquakes. Since we know the exact days and intensities for these 5 earthquakes we can compute the optimal value of K for anomaly detection methods and generate our network for ANN (since many studies have investigated these earthquakes using similar anomaly detection methods we could also use their results regarding the optimal value for K). Now that the K value and our network are generated we can move to the next step, the 6th earthquake. It is true that we do not have the coordinates of the epicentre for this earthquake; we do know the locations of all the faults in the country (on greater scale the world). We can use data regarding every fault to test them in anomaly detection method and ANN to see if any anomalies can be detected in them and if they do what intensity these anomalies can cause. Since the fault correspond to the 6th earthquake is one of the faults in the country we ultimately see anomalies regarding the data from this fault which indicates the possibility in occurrence of an earthquake near the area with the intensity computed from ANN.

General comments

1-Fault identification: often, a "fault systems" rather than a single fault, is activated during strong EQs. How the method can handle this? Not clear. In addition, the identification and selection of the fault is, even in this work, carried out aftermath, when the epicentre position is well known: how can manage this in terms of EQ forecast?

In cases that multiple faults are engaged and a "fault systems" is activated, buffers can still be made around the multiple faults but it is highly recommended to consider only active faults to reduce the number of faults and test each fault individually to see which one has the similar pattern to the rest of the earthquakes.

2-LST anomaly identification: to remove natural/observational noise, authors consider ". . .a previous year with no strong seismic activities. . ." and subtract this "linear function" from the data. In my opinion, 1 year is not enough to be considered as representative of the actual "normal" conditions of the area. A single year, in fact, can be affected by other (e.g. meteorological) forcing factors, limiting its representativeness for a "normal" year and significantly impacting in thermal anomaly identification.

It is true that using only a linear function and data regarding the previous year cannot entirely removes natural/observational noises it does remove the seasonal changes, which contains majority of these noises. Moreover, since LST's changes depend on various factors it is almost impossible to remove these noises completely.

3-coefficients k: to detect a thermal anomaly, authors apply two different methods, both requiring a threshold test based on a "k" coefficient that seems determined in a totally arbitrary way. K values determination, as well as their possible variability and dependence on different environmental/observational/geographical conditions should be better justified and assessed. Additionally, it is not completely clear how the average and sigma are computed.

As mentioned, we can compute the optimal value of K using data regarding the other 5 earthquakes. As for the average and sigma, we will insert their equations in the future manuscript accordingly.

4-Buffer radius: authors consider LST mean computed in buffer radius from 1 to 20 km. Why limiting R < 20km? It is well known that the "preparation zone" of a strong EQ could be as large as the Dobrovolsky theory (Dobrovolsky et al. 1979), i.e R = 100,43M (with M= magnitude). In particular, for M=6 a radius of about 400 km can be expected for the EQ preparation zone. Moreover, if thermal anomalies are related to fault degassing, gases (e.g. $CO_2$) might be spread in large areas, also depending on meteorological conditions (e.g. wind intensity and direction) and/or local topography. Therefore, the limitation at 20 km should be better explained and scientifically justified.

It is true that the earthquake's preparation zone is much farther than 20km but in distant zones, each earthquake follows a different pattern and using them in ANN and anomaly detection method will result in very weak network and lower the outcome accuracy.

5-How ANN works to estimate EQ intensity? Have the authors trained the network using the same dataset they used for results or an independent one?

I hope my explanation in the beginning of this letter provides the necessary information regarding this issue.

Specific comments

Figure 3: what do the black dots mean?

The black dots in figure 3 represent the LST mean value in each day and buffer zone. curve fitting was used to show a smoother curve in diagram.(the smooth fitted curve is just for better showing the diagram and the exact LST mean was used in any algorithm mentioned in this study)

-It is not clear how the fault distance map is used in the process. Please clarify.

-Not clear how the LST mean is computed in buffer radius. In particular, please better explain the following sentence: "It should be noted that width of each buffer is only 1 km and R is the buffer radius (distance) from the related fault."

These two comments are connected. The LST mean is computed in each buffer radius using the fault distance map. Since pixels in same buffer zones have the same value in the fault distance map, we could use it to average the LSTs in each buffer zone. "It should be noted that width of each buffer is only 1 km and R is the buffer radius (distance) from the related fault" this phrase means each buffer is subtracted from its next buffer. For example, the buffer with 4 km radius consists of only pixels in the range of 3-4 km.

-Lines 204-207: authors assert that anomaly far from the epicentre are not used for ANN. Thus, authors are using epicentre information for filtering data, but epicentre position is only available after EQ occurred, so this study appears as a retrospective analysis as well, and cannot be used for forecasting

I hope my explanation in the beginning of this letter provides the necessary information regarding this issue.

-Lines 230-231: "Although these anomalies are not as strong as the anomalies detected near the time of the earthquake, they seem to be related to some seismic activities rather than being a false alarm": how authors can assert this? What is the scientific basis for asserting this?

Some earthquakes (aftershocks) was reported around the same days these anomalies were detected but their intensities were not even close to the main earthquakes (MS$\leq$4). That's why we could not be certain about these anomalies on what have caused them but perhaps using the word "false alarm" is a strong word so we will change it in the future manuscript.

-Lines 233-234: ..." Many anomalies detected by this method are related to the earthquake...". Again, how authors can affirm this?

What we meant was that these anomalies are most likely be related to the earthquake and can be used in anomaly detections and ANN since they appeared around the day of the earthquakes and have similar pattern in most of the earthquakes.

-Lines 244-245: "ANN results also show high correlations between thermal anomaly data and 245 the earthquakes intensity." What exactly does this sentence mean?

This sentence means that the stronger the earthquake the greater the difference LST between the detected anomaly and its surroundings both in day and distance. In other word, a strong earthquake not only has a greater anomaly on the day of the earthquake during the investigation days but also has a greater anomaly in the nearest buffer to the fault on the day of the earthquake.

Cited reference Dobrovolsky I P, Zubkov S I and Myachkin V I, 1979 Estimation of the size of the earthquake preparation zones Pure Appl. Geophys. 117 1025–44.

Please also note the supplement to this comment:
https://nhess.copernicus.org/preprints/nhess-2020-391/nhess-2020-391-AC1-supplement.pdf

---

## Referee Comment (RC2) · Anonymous Referee #2 · 7 May 2021

The current manuscript presents a new method for correlating strong earthquakes with thermal anomalies using LST records for the purposes of earthquake prediction and/or forecasting. The subject is within the scope of the journal but even the idea is interesting, the manuscript at its current form cannot be published due to the following major issues:

1) In literature, there are many proposed methods for strong earthquake prediction based on observed singularities of physical quantities. The non-prevalence of a specific one (or at least a limited set of them) means that the detection of the aforementioned singularities in not enough to have an operationally complete method. What is usually

missed is the performance of each proposed method during the periods or situations where there is no intense activity (or at periods with mild activity). This is linked to what we called "false alarm avoidance". the current manuscript suffers from this lack also: the authors did not present the results of their proposed method in the periods where no strong events existed in order to compare both results. If this is not done then this study is incomplete

2) In the case that authors can successfully implement the previous suggestions, the authors must justify the selection of some crucial parameters of their method such as: a) how the k coefficient was selected b) how the value of radius R was selected? based on topographic criteria or due to some signal processing axioms? in order to catch up a very common answer, the significance of the proposed method could not be based on "empirical selection" and thus must justified in a solid framework

3) the authors claim the use of an ANN in order to estimate the EQ intensity. this could be acceptable only if the authors provide details about the implementation of the ANN that they used (topology, comparison to relevant implementations, performance, training data set, criteria for selection of training and evaluation data sets, scoring of individual runs, computing requirements)

For the above reasons i suggest a major revision of the current manuscript

---

## Author Comment (AC2) · 5 Jun 2021

We appreciate the time and effort that you have dedicated to providing your valuable feedback and your insightful comments on our manuscript. We have been able to incorporate changes to reflect most of the suggestions you have provided and will be accessible in our next uploading manuscript.

1) In literature, there are many proposed methods for strong earthquake prediction based on observed singularities of physical quantities. The non-prevalence of a specific one (or at least a limited set of them) means that the detection of the aforementioned

singularities in not enough to have an operationally complete method. What is usually missed is the performance of each proposed method during the periods or situations where there is no intense activity (or at periods with mild activity). This is linked to what we called "false alarm avoidance". The current manuscript suffers from this lack also: the authors did not present the results of their proposed method in the periods where no strong events existed in order to compare both results. If this is not done then this study is incomplete We can use the method on the previous year data for each earthquake which no seismic activities was reported to show and compare their results with the results generated from the year of the earthquake in the future manuscript.

2) In the case that authors can successfully implement the previous suggestions, the authors must justify the selection of some crucial parameters of their method such as: a) how the k coefficient was selected b) how the value of radius R was selected? Based on topographic criteria or due to some signal processing axioms? In order to catch up a very common answer, the significance of the proposed method could not be based on "empirical selection" and thus must justified in a solid framework.

a) Since for each earthquake that we investigate the other 5 earthquakes are considered to have already occurred and their details are known. Using the information regarding these 5 earthquakes we have found the optimal value of K for anomaly detection methods for our region. We have also compare them with similar studies that investigated same earthquakes.

b) Since we are using MODIS products of LST with spatial resolution of 1km the interval for R is set to this same amount so that no transformation had to be done. As we know strong earthquakes have large area of effect and many studies used MODIS data for their investigating we can assume this amount to be sufficient for our study. As for the range of R we have to consider that not only by increasing this parameter we can see the less effects caused by the earthquake on the outlying layers we also put several different landcovers with various temperature in one layer. This will result inaccuracy in later processing. Using different parameters such as Variance can help us better find

the optimal maximum range for R.

3) The authors claim the use of an ANN in order to estimate the EQ intensity. This could be acceptable only if the authors provide details about the implementation of the ANN that they used (topology, comparison to relevant implementations, performance, training data set, criteria for selection of training and evaluation data sets, scoring of individual runs, computing requirements).

We will add more detail results of our ANN such as its performance, comparison to relevant implementations, training data set and criteria for selection of training and evaluation data sets in the future manuscript.